# THE FORWARD-BACKWARD EMBEDDING OF DIRECTED GRAPHS

## ABSTRACT

We introduce a novel embedding of directed graphs derived from the singular value decomposition (SVD) of the normalized adjacency matrix. Specifically, we show that, after proper normalization of the singular vectors, the distances between vectors in the embedding space are proportional to the mean commute times between the corresponding nodes by a forward-backward random walk in the graph, which follows the edges alternately in forward and backward directions. In particular, two nodes having many common successors in the graph tend to be represented by close vectors in the embedding space. More formally, we prove that our representation of the graph is equivalent to the spectral embedding of some co-citation graph, where nodes are linked with respect to their common set of successors in the original graph. The interest of our approach is that it does *not* require to build this co-citation graph, which is typically much denser than the original graph. Experiments on real datasets show the efficiency of the approach.

## 1 INTRODUCTION

Learning from data structured as a graph most often requires to embed the nodes of the graph in some Euclidian space (Yan et al. (2007); Grover and Leskovec (2016); Bronstein et al. (2017)). The ability to learn then critically depends on this embedding, that must incorporate the "geometry" of the graph in some sense. A classical embedding of *undirected* graphs is based on the spectral decomposition of the Laplacian (Belkin and Niyogi (2003)); after proper normalization, the distances between vectors in the embedding space are proportional to the mean commute times of the corresponding nodes by a random walk in the graph, making this embedding meaningful and easy to interpret (Qiu and Hancock (2007); Fouss et al. (2007)). This result is not applicable to *directed* graphs.

In this paper, we show that, after proper normalization, the embedding derived from the singular value decomposition (SVD) of the normalized adjacency matrix of a directed graph, as considered in (Dhillon (2001); Rohe et al. (2012)), can also be interpreted in terms of a random walk in the graph. This random walk is particular in the sense that edges are followed alternately in forward and backward directions. We call it the *forward-backward* random walk, and the corresponding embedding as the *forward-backward* embedding. As for undirected graphs, the distances between vectors in the embedding space are proportional to the mean commute times between the corresponding nodes by the random walk. We show in fact that the forward-backward embedding of a graph is equivalent to the spectral embedding of some co-citation graph. The interest of our approach is that it does *not* require to build this co-citation graph, which is typically much denser than the original graph.

The rest of the paper is organized as follows. We first present the related work. We then introduce the Laplacian matrix of a directed graph and the notion of co-citation graph. The embedding, and its interpretation in terms of the forward-backward random walk, are presented in Sections 5 and 6, respectively, under the assumption that the co-citation graph is connected; this assumption is relaxed in Section 7. The link between our embedding and the spectral embedding is explained in Section 8. The practically interesting case of bipartite graphs, that may be viewed as particular instances of directed graphs, is considered in Section 9. We conclude the paper by experiments on real datasets in Section 10.

We denote by 1 the vectors of ones of appropriate dimension. For any square matrix $M$, $M^+$ is the Moore-Penrose inverse of $M$ and $d(M)$ is the diagonal matrix which has the same diagonal as $M$.

## 2 RELATED WORK

While the spectral embedding of undirected graphs, and its interpretation in terms of random walks, is very well understood (see Lovász (1993); Snell and Doyle (2000); Qiu and Hancock (2007); Fouss et al. (2007); Luxburg (2007)), the results do not easily extend to directed graphs. The main reason is that the Laplacian matrix is no longer symmetric and thus has complex eigenvalues in general. In (Li and Zhang (2010); Boley et al. (2011)), it is shown that the mean commute times of the random walk can be expressed in terms of the pseudo-inverse of some proper Laplacian matrix, extending the results known for undirected graphs (Qiu and Hancock (2007)). The considered Laplacian relies on the stationary distribution of the random walk, which is not explicit in general, and on the assumption that the graph is strongly connected. Moreover, no embedding of the graph (e.g., using the SVD of the Laplacian, as suggested in Li and Zhang (2010)) is known to be directly related to the mean commute times of the random walk, as for undirected graphs.

The embedding of directed graphs proposed in (Dhillon (2001); Rohe et al. (2012)) relies on the SVD of the normalized adjacency matrix. Our main contribution is a proper normalization of the singular vectors so that the distances in the embedding space can be interpreted in terms of mean commute times of the forward-backward random walk. In particular, our embedding is *not* an extension of the usual spectral embedding of undirected graphs, a point that is addressed specifically in Section 8. The idea of a random walk alternating between forward and backward steps is in the spirit of HITS (Kleinberg et al. (1999)), an algorithm proposed to rank Web pages in terms of their relative importances as so-called hubs and authorities.

Various other embedding techniques rely on random walks in the graph (see Perrault-Joncas and Meila (2011); Cai et al. (2018); Goyal and Ferrara (2018); Qiu et al. (2018) and references therein). These include DeepWalk (Perozzi et al. (2014)) and node2vec (Grover and Leskovec (2016)), inspired by Natural Language Processing techniques, where the representation of the graph is learned from (simulated) random walks in the graph. Unlike the embedding proposed in the present paper, there is no direct interpretation of the distances in the embedding space in terms of random walk in the graph.

## 3 LAPLACIAN MATRIX

Let $G$ be some directed graph and $A$ its adjacency matrix. For convenience, we present the results for unweighted graphs (binary matrices $A$), but the results readily apply to weighted graphs (non-negative matrices $A$). We denote by $n$ the number of nodes and by $m = 1^T A 1$ the number of edges (the total weight of edges for a weighted graph). Unless otherwise specified, we assume that there are neither sources nor sinks in the graph, i.e., each node has a positive indegree and a positive outdegree. The extension to general graphs is considered in Section 7. We refer to the (normalized) Laplacian as:

$$L = I - D_{\text{out}}^{-\frac{1}{2}} A D_{\text{in}}^{-\frac{1}{2}}, \tag{1}$$

where $D_{\text{out}} = \text{diag}(A1)$ and $D_{\text{in}} = \text{diag}(A^T 1)$ are the diagonal matrices of outdegrees and indegrees, respectively. We refer to the matrix

$$D_{\text{out}}^{-\frac{1}{2}} A D_{\text{in}}^{-\frac{1}{2}}$$

as the normalized adjacency matrix. Observe that the matrix $L^T$ is the Laplacian of the reverse graph $G^-$ (same graph as $G$ but with reverse edges), with adjacency matrix $A^T$. In particular, the Laplacian $L$ is symmetric if and only if $A$ is symmetric, in which case we recover the usual normalized Laplacian of undirected graphs, namely $L = I - D^{-\frac{1}{2}} A D^{-\frac{1}{2}}$ with $D = D_{\text{out}} = D_{\text{in}}$.

## 4 Co-citation graph

The co-citation graph associated with $G$ is the graph with adjacency matrix $AA^T$. This is a weighted, undirected graph where the weight between nodes $i$ and $j$ is the number of common successors of $i$ and $j$.

We refer to the *normalized* co-citation graph associated with $G$ as the graph with adjacency matrix $\bar{A} = AD_{\text{in}}^{-1}A^T$. This undirected graph has the same edges as the co-citation graph but the weight associated to any common successor of $i$ and $j$ is now normalized by its indegree. For instance, two articles of Wikipedia, say $i, j$, pointing to the same other article, say $k$, will be all the more similar (higher weight) in the corresponding normalized co-citation graph as article $k$ is referenced by fewer other articles (lower in-degree) in the original graph. The node weights in the normalized co-citation graph (total weights of incident edges) are equal to the out-degrees in the graph $G$:

$$\bar{A}1 = AD_{\text{in}}^{-1}A^T1 = A1. \tag{2}$$

The normalized Laplacian of the normalized co-citation graph is:

$$\bar{L} = I - \bar{D}^{-\frac{1}{2}}\bar{A}\bar{D}^{-\frac{1}{2}},$$

where $\bar{D} = \text{diag}(\bar{A}1)$ is the diagonal matrix of node weights in the normalized co-citation graph. Observe that $\bar{D} = D_{\text{out}}$ in view of (2).

**Proposition 1** *We have $I - \bar{L} = (I - L)(I - L)^T$.*

*Proof.* This follows from $I - \bar{L} = \bar{D}^{-\frac{1}{2}}\bar{A}\bar{D}^{-\frac{1}{2}} = D_{\text{out}}^{-\frac{1}{2}}AD_{\text{in}}^{-1}A^TD_{\text{out}}^{-\frac{1}{2}} = (I - L)(I - L)^T$. $\square$

## 5 Forward-backward embedding

Consider a singular value decomposition (SVD) of the normalized adjacency matrix[1],

$$D_{\text{out}}^{-\frac{1}{2}}AD_{\text{in}}^{-\frac{1}{2}} = U\Sigma V^T, \tag{3}$$

where $\Sigma = \text{diag}(\sigma_1, \ldots, \sigma_n)$ with $\sigma_1 \geq \ldots \geq \sigma_n \geq 0$ and $U^TU = V^TV = I$. We get

$$L = I - U\Sigma V^T,$$

and by Proposition 1,

$$\bar{L} = U(I - \Sigma^2)U^T. \tag{4}$$

Thus the normalized Laplacian of the normalized co-citation graph has eigenvalues $1 - \sigma_1^2 \leq \ldots \leq 1 - \sigma_n^2$ with corresponding unitary matrix of eigenvectors $U$. These eigenvalues are non-negative, the multiplicity of the eigenvalue 0 being the number of connected components of this graph (Luxburg (2007)). We deduce that $\sigma_1 = \ldots = \sigma_K = 1 > \sigma_{K+1}$, where $K$ is the number of connected components of the co-citation graph.

Assume that the co-citation graph is connected. The general case is considered in Section 7. Then $\sigma_1 = 1 > \sigma_2$. Let $\Gamma = (I - \Sigma^2)^{\frac{1}{2}}$ and define:

$$X = \Gamma^+U^TD_{\text{out}}^{-\frac{1}{2}}. \tag{5}$$

The columns $x_1, \ldots, x_n$ of $X$ define a representation of the graph in $\mathbb{R}^n$. Specifically, each node $i$ is represented by the vector $x_i \in \mathbb{R}^n$. Observe that the first coordinate of each vector $x_i$ is null so that at most $n - 1$ coordinates are informative. The Euclidian distances between these vectors are entirely defined by the Gram matrix of $X$, which is related to the pseudo-inverse of the normalized Laplacian of the normalized co-citation graph,

$$\bar{L}^+ = U(I - \Sigma^2)^+U^T.$$

---

[1] A naive approach to the SVD of a matrix $M$ is based on the spectral decomposition of the matrix $MM^T$; in our case, this would be equivalent to build the co-citation graph, which is typically much denser than the original graph. This is why we use a proper implementation of the SVD that works directly on the matrix $M$, see Halko et al. (2011).

In view of (5),

$$X^T X = D_{\text{out}}^{-\frac{1}{2}} \bar{L}^+ D_{\text{out}}^{-\frac{1}{2}}.$$

We show in the following section that the square distance between $x_i$ and $x_j$ is proportional to the mean commute time between nodes $i$ and $j$ of the *forward-backward* random walk in the graph $G$.

In practice, the graph is embedded in some vector space of dimension $d$, chosen much lower than $n$. In this case, only the $d$ leading singular vectors are considered, i.e., those associated with the singular values $\sigma_1, \ldots, \sigma_d$ (or the $d + 1$ leading singular vectors is the first vector, which is not informative, is skipped).

## 6   Forward-backward random walk

Consider a random walk in the original graph $G$ where edges are followed in forward and backward directions alternately. Specifically, from node $i$, a successor of $i$ is chosen uniformly at random, say node $j$; then a predecessor of $j$ (possibly $i$) is chosen uniformly at random, say $k$. Thus each jump of the random walk, here from $i$ to $k$, involves two moves, here from $i$ to $j$ (forward) then from $j$ to $k$ (backward). The successive nodes $X_0, X_1, \ldots$ visited by this forward-backward random walk form a Markov chain on the set of nodes with transition matrix $\bar{P} = D_{\text{out}}^{-1} A D_{\text{in}}^{-1} A^T$, that is, the probability of a jump from $i$ to $j$ is:

$$\bar{P}_{ij} = \sum_{k=1}^{n} \frac{A_{ik} A_{jk}}{(D_{\text{out}} 1)_i (D_{\text{in}} 1)_k}.$$

Equivalently, this Markov chain corresponds to a standard random walk in the normalized co-citation graph, in view of the equality $\bar{P} = \bar{D}^{-1} \bar{A}$.

Let $H$ be the matrix of mean hitting times, i.e., $H_{ij}$ is the mean hitting time of node $j$ from node $i$. We have $H_{ii} = 0$ and for all $i \neq j$,

$$H_{ij} = 1 + \sum_{k=1}^{n} \bar{P}_{ik} H_{kj},$$

so that the matrix $(I - \bar{P})H - 11^T$ is diagonal. The following result shows that $H$ is directly related to the Gram matrix of $X$. This is a consequence of the fact that the forward-backward random walk in the graph $G$ corresponds to a regular random walk in the normalized co-citation graph. A proof is provided for the sake of completeness.

**Proposition 2** *We have:*

$$H = m(11^T d(X^T X) - X^T X). \tag{6}$$

*Proof.* Since the matrix $\bar{P}$ is stochastic, the matrix $H$ defined by (6) satisfies:

$$(I - \bar{P})H = -m(I - \bar{P})X^T X.$$

Since $I - \bar{P} = \bar{D}^{-\frac{1}{2}} \bar{L} \bar{D}^{\frac{1}{2}}$, we get:

$$(I - \bar{P})H = -m\bar{D}^{-\frac{1}{2}} \bar{L} \bar{L}^+ \bar{D}^{-\frac{1}{2}},$$

$$= -m\bar{D}^{-\frac{1}{2}}(I - \bar{D}^{\frac{1}{2}} \frac{11^T}{m} \bar{D}^{\frac{1}{2}})\bar{D}^{-\frac{1}{2}},$$

$$= -m\bar{D}^{-1} + 11^T,$$

where the second equality comes from the fact that $\bar{D}^{\frac{1}{2}} 1/\sqrt{m}$ is the unitary eigenvector of the normalized Laplacian $\bar{L}$ for the eigenvalue 0. In particular, the matrix $(I - \bar{P})H - 11^T$ is diagonal. Moreover, the matrix $H$ is zero-diagonal. The proof follows from the fact that the matrix of mean hitting times is uniquely defined by these two properties.  $\square$

Let $C = H + H^T$ be the matrix of mean commute times, i.e., $C_{ij}$ is the mean commute time between nodes $i$ and $j$. It follows directly from Proposition 2 that $H_{ij} = m(x_j - x_i)^T x_j$ and $H_{ji} = m(x_i - x_j)^T x_i$, so that $C_{ij} = m||x_i - x_j||^2$, i.e., the mean commute time between nodes $i$ and $j$ is proportional to the square Euclidian distances between $x_i$ and $x_j$ in the embedding space.

Now let $\pi$ be the stationary distribution of the random walk. The mean hitting time of node $i$ in steady state is given by:

$$h_i = \sum_{i=1}^{n} \pi_j H_{ji}.$$

Since $\pi = \bar{D}1/m$, we have $X\pi = 0$ and, in view of Proposition 2, $h_i = m||x_i||^2$: the mean hitting time of node $i$ is proportional to the square norm $x_i$ in the embedding space.

Finally, we have $mx_i^T x_j = h_j - H_{ij} = h_i - H_{ji}$ so that:

$$\cos(x_i, x_j) = \frac{h_i + h_j - C_{ij}}{2\sqrt{h_i h_j}}.$$

The cosine similarity between $x_i$ and $x_j$ can thus be interpreted in terms of mean commute times between nodes $i$ and $j$ *relative* to the mean hitting times $h_i, h_j$. In particular, the vectors $x_i$ and $x_j$ are close in terms of cosine similarity if the mean commute time $C_{ij}$ is small compared to $h_i + h_j$. This is equivalent to consider the embedding where each vector $x_i$ is normalized by its Euclidian norm $||x_i||$.

## 7 GENERAL GRAPHS

In this section, we relax the assumptions that the out-degrees and in-degrees are positive and that the co-citation graph is connected. Let $C^{(1)}, \ldots, C^{(K)}$ be the connected components of the co-citation graph. These sets form a partition of the set of nodes of the original graph $G$. Observe that each sink of the graph $G$ is isolated in the co-citation graph so that the forward-backward random walk starting from such a node is not defined. In the following, we consider any connected component $C^{(k)}$ not reduced to a single node. The forward-backward random walk starting from any node in $C^{(k)}$ is well defined and corresponds to an irreducible Markov chain on $C^{(k)}$. Let $A^{(k)}$ be the restriction of $A$ to its rows indexed by $C^{(k)}$, that is, $A_{ij}^{(k)}$ is defined for each $i \in C^{(k)}$ and any $j$ and equal to $A_{ij}$. This is a matrix of dimension $n^{(k)} \times n$ where $n^{(k)}$ is the number of nodes in the connected component $C^{(k)}$.

Consider a singular value decomposition of the matrix :

$$D_{\text{out}}^{(k)}{}^{-\frac{1}{2}} A^{(k)} (D_{\text{in}}^+)^{\frac{1}{2}} = U^{(k)} \Sigma^{(k)} V^{(k)T},$$

where $D_{\text{out}}^{(k)} = \text{diag}(A^{(k)}1)$. Observe that the diagonal entries of $D_{\text{out}}^{(k)}$ are positive. Let $\Gamma^{(k)} = (I - \Sigma^{(k)2})^{\frac{1}{2}}$ and define:

$$X^{(k)} = \Gamma^{(k)+} U^{(k)T} D_{\text{out}}^{(k)}{}^{-\frac{1}{2}}.$$

This is the forward-backward embedding of the nodes of $C^{(k)}$ in $\mathbb{R}^{n^{(k)}}$. By the same argument as before, the square Euclidian distances between vectors in the embedding space are proportional to the mean commute times of the forward-backward random walk in the graph $G$, starting from any node in $C^{(k)}$.

## 8 LINK WITH SPECTRAL EMBEDDING

Any undirected graph $G$ can be viewed as a directed graph with edges in both directions. The square Euclidian distances between vectors in the embedding space then correspond to mean commute times of a *two-hop* random walk in the graph $G$. In particular, the proposed embedding differs from the usual spectral embedding whose geometry is related to a regular *one-hop* random walk in the graph $G$ (Qiu and Hancock (2007)).

Specifically, the normalized Laplacian $L = I - D^{-\frac{1}{2}}AD^{-\frac{1}{2}}$, with $D = D_{\text{out}} = D_{\text{in}}$, is symmetric and thus admits a spectral decomposition of the form:

$$L = W\Lambda W^T, \tag{7}$$

where $\Lambda = \text{diag}(\lambda_1, \ldots, \lambda_n)$, with $\lambda_1 \leq \ldots \leq \lambda_n$ and $W^T W = I$. Since $I - L$ has the same eigenvalues as $P = D^{-1}A$, the transition matrix of the regular random walk in graph $G$, we have $0 \leq \lambda_1 \leq \ldots \leq \lambda_n \leq 2$, with $\lambda_n = 2$ if and only if the graph $G$ is bipartite. Define:

$$Z = (\Lambda^{\frac{1}{2}})^+ W^T D^{-\frac{1}{2}}. \tag{8}$$

The columns $z_1, \ldots, z_n$ of $Z$ provide a spectral embedding of the graph $G$, such that the square distance between $z_i$ and $z_j$ is proportional to the mean commute time between nodes $i$ and $j$ of a regular random walk in the graph $G$.

Now it follows from Proposition 1 that

$$I - \bar{L} = (I - L)^2 = W(I - \Lambda)^2 W^T.$$

Let $\phi$ be a permutation of $\{1, \ldots, n\}$ such that:

$$(1 - \lambda_{\phi(1)})^2 \geq \ldots \geq (1 - \lambda_{\phi(n)})^2.$$

Since

$$I - \bar{L} = U\Sigma^2 U^T,$$

we get

$$\sigma_1^2 = (1 - \lambda_{\phi(1)})^2, \ldots, \sigma_n^2 = (1 - \lambda_{\phi(n)})^2,$$

and $U = W_\phi$ (permutation $\phi$ of the columns of $W$) whenever all singular values of $I - L$ are distinct (otherwise, the equality holds up to a rotation of the singular vectors associated with the same singular values). We deduce that the spectral embedding $Z$ is the same as the forward-backward embedding $X$, up to a permutation, possible rotations, and a renormalization ($k$-th column of $U$ normalized by $\sqrt{\lambda_k}$ instead of $\sqrt{1 - (1 - \lambda_k)^2} = \sqrt{\lambda_k(2 - \lambda_k)}$). Observe that the difference may be significant in the presence of eigenvalues close to 2, where the graph tends to have a bipartite structure. Moreover, the permutation $\phi$ implies that the order of the singular vectors of $I - L$ (used in the forward-backward embedding $X$) is not that of the eigenvectors of $L$ (used in the spectral embedding $Z$), so that the corresponding embeddings induced by the $d$ leading singular vectors for some $d << n$ may be very different.

## 9 Bipartite graphs

A bipartite graph $G$ with two sets of nodes $\mathcal{N}_1$ and $\mathcal{N}_2$ can be viewed as a directed graph with an edge from $i_1 \in \mathcal{N}_1$ and $i_2 \in \mathcal{N}_2$ for any edge between $i_1$ and $i_2$ in $G$. The forward-backward embedding of this directed graph provides a representation of the nodes $\mathcal{N}_1$ in $\mathbb{R}^{n_1}$, where $n_1 = |\mathcal{N}_1|$ is the numbers of nodes in $\mathcal{N}_1$ (the embedding of the nodes $\mathcal{N}_2$ is obtained by reversing the edges).

It is then more convenient to work with the biadjacency matrix $B$ of $G$, of dimension $n_1 \times n_2$ with $n_1 = |\mathcal{N}_1|$ and $n_2 = |\mathcal{N}_2|$: $B_{i_1,i_2} = 1$ if and only if there is an edge between node $i_1 \in \mathcal{N}_1$ and $i_2 \in \mathcal{N}_2$. Let $D_1 = \text{diag}(B1)$ and $D_2 = \text{diag}(B^T 1)$ be the diagonal matrices of the degrees of each part. We assume that the diagonal entries of $D_1$ and $D_2$ are positive (equivalently, there is no node of null degree in $G$). Consider a singular value decomposition of the form:

$$D_1^{-\frac{1}{2}} B D_2^{-\frac{1}{2}} = U_1 \Sigma U_2^T,$$

where $\Sigma$ is a $n \times n$ non-negative diagonal matrix, with $n = \min(n_1, n_2)$. Let $\Gamma = (I - \Sigma^2)^{\frac{1}{2}}$. The forward-backward embeddings of $\mathcal{N}_1$ and $\mathcal{N}_2$ are then given by:

$$X_1 = \Gamma^+ U_1^T D_1^{-\frac{1}{2}} \quad \text{and} \quad X_2 = \Gamma^+ U_2^T D_2^{-\frac{1}{2}}.$$

## 10 Experiments

In this section, we assess the quality of the forward-backward embedding on a standard clustering task. Specifically, we apply k-means clustering to various embeddings of graphs of the Koblenz Network Collection[1], a rich collection of more than 250 graphs (Kunegis (2013)).

---

[1] http://konect.uni-koblenz.de

Due to the absence of ground-truth clusters, we measure the quality of the clustering through its modularity in the normalized co-citation graph:

$$Q = \frac{1}{m} \sum_{i,j=1}^{n} \left( \bar{A}_{ij} - \frac{\bar{d}_i \bar{d}_j}{m} \right) \delta_{c_i, c_j},$$

where $\bar{d} = \bar{A}1$ is the vector of degrees in the normalized co-citation graph, $c_i, c_j$ are the clusters of nodes $i, j$ and $\delta$ is the Kronecker delta (Newman and Girvan (2004)). This quantity lies in the interval $[-1, 1]$, the value $Q = 0$ corresponding to the trivial clustering with a single cluster; the higher the modularity $Q$, the better the clustering. The modularity can be written with respect to the adjacency matrix of the original graph as:

$$Q = \frac{1}{m} \sum_{i,j=1}^{n} 1_i^T A \left( D_{\text{in}}^{-1} - \frac{11^T}{m} \right) A^T 1_j \delta_{c_i, c_j},$$

where $1_i, 1_j$ are the unit vectors on components $i, j$. In particular, the computation of the modularity does not require that of $\bar{A}$, the adjacency matrix of the normalized co-citation graph; it can be directly evaluated in $O(m)$ operations by matrix-vector multiplications from the adjacency matrix $A$.

The considered embeddings are the following:

- Identity embedding (Id): a baseline where the k-means algorithm is directly applied to the adjacency matrix $A$ (or to the biadjacency matrix $B$ for bipartite graphs);
- Dhillon co-clustering (Dh), for bipartite graphs only, based on the SVD of the normalized adjacency matrix, without normalization (Dhillon (2001));
- Laplacian Eigenmaps (LE): the spectral decomposition of the graph, considered as undirected, as described in Belkin and Niyogi (2003);
- Forward-Backward embedding (FB): the embedding (5), normalized to get unitary vectors, as described at the end of Section 6.

The Python code used for the experiments is available as a Jupyter notebook[2], making the experiments fully reproducible.

Tables 1 and 2 show the modularity scores and running times of each embedding for 30 directed graphs and 20 bipartite graphs of the Konect collection[1], each graph being identified through its code in the collection. We have selected these representative sets of graphs because of space constraints but the Jupyter notebook available online[2] can be run on *all* graphs of the Konect collection. The graphs are considered as simple and unweighted. We give in Tables 1 and 2 the size of each graph, which may be lower than that announced in the Konect collection if the original graph is a multi-graph, as we count each edge only once. The running times are for a PC equipped with a AMD Ryzen Threadripper 1950X processor and a RAM of 32GB. Running times exceeding 1000s trigger a time-out[2]. The dimension of the embedding is $d = 10$ for all embeddings except Id where we take the full matrix. The clustering algorithm is k-means++ with 10 clusters Arthur and Vassilvitskii (2007).

The results show that FB is extremely fast and provide generally a much better embedding than Dh and LE with respect to the considered clustering task. They confirm experimentally the relevance of the theoretical results shown in this paper, namely the interpretation of distances in the embedding space in terms of a forward-backward random walk in the graph, or a regular random walk in the normalized co-citation graph. Although these theoretical results apply in principle to the full embedding only, i.e., $d = n$, the experimental results tend to show that the FB embedding also captures the "geometry" of the normalized co-citation graph in low dimension, here $d = 10$. Again, our approach does not require the construction of this graph, whose structure is directly captured by the SVD.

---

[2]`https://github.com/tbonald/directed`
[2]We have also tested the Node2Vec embedding Grover and Leskovec (2016), available on the Jupyter notebook, but this leads to a timeout for almost all graphs of the Konect collection on the PC used in the experiments.

| Dataset | Size (#edges) | Id | LE | FB |
|---|---|---|---|---|
| MS | 6 207 | 0.26 (< 1s) | 0.01 (3s) | **0.37** (< 1s) |
| Mg | 19 025 | 0.17 (< 1s) | 0.00 (< 1s) | **0.49** (< 1s) |
| AD | 51 127 | **0.16** (< 1s) | 0.04 (5s) | 0.08 (< 1s) |
| CH | 65 053 | 0.07 (< 1s) | 0.00 (6s) | **0.25** (< 1s) |
| DG | 87 627 | 0.13 (< 1s) | 0.01 (4s) | **0.26** (1s) |
| CC | 91 500 | 0.09 (< 1s) | 0.45 (4s) | **0.60** (2s) |
| SD | 140 778 | 0.09 (< 1s) | 0.00 (15s) | **0.27** (2s) |
| GN | 147 892 | 0.00 (< 1s) | 0.00 (15s) | **0.46** (2s) |
| DJ | 150 985 | 0.30 (< 1s) | **0.37** (1s) | 0.11 |
| LX | 213 954 | **0.20** (< 1s) | 0.00 (9s) | 0.17 |
| EA | 312 342 | 0.00 (< 1s) | 0.18 (3s) | **0.19** (2s) |
| PHc | 421 578 | 0.07 (< 1s) | 0.00 (33s) | **0.60** (2s) |
| ES | 508 837 | 0.13 (1s) | 0.00 (14s) | **0.14** (3s) |
| Ow | 876 933 | 0.01 (< 1s) | 0.01 (202s) | **0.51** (2s) |
| EN | 1 148 072 | 0.11 (< 1s) | 0.00 (107s) | **0.44** (2s) |
| ND | 1 497 134 | 0.38 (1s) | TO | **0.79** (5s) |
| DF | 1 731 653 | **0.19** (1s) | 0.01 (161s) | 0.14 (5s) |
| CS | 1 751 463 | 0.21 (2s) | 0.01 (295s) | **0.55** (7s) |
| SF | 2 312 497 | 0.45 (3s) | TO | **0.79** (6s) |
| BAr | 3 284 387 | 0.52 (3s) | TO | **0.68** (7s) |
| Am | 3 387 388 | 0.00 (3s) | **0.65** (417s) | 0.59 (8s) |
| GO | 5 105 039 | 0.02 (4s) | TO | **0.72** (12s) |
| BS | 7 600 595 | 0.53 (4s) | TO | **0.79** (10s) |
| DB | 13 820 853 | **0.51** (15s) | TO | 0.45 (44s) |
| PC | 16 518 947 | 0.22 (15s) | TO | **0.45** (64s) |
| LI | 17 359 346 | 0.12 (18s) | 0.24 (26s) | **0.37** (19s) |
| HUr | 18 854 882 | 0.25 (11s) | TO | **0.31** (42s) |
| PL | 30 622 564 | 0.00 (15s) | 0.39 (229s) | **0.51** (56s) |
| FL | 33 140 017 | 0.18 (14s) | TO | **0.45** (47s) |
| LJ | 68 475 391 | 0.00 (43s) | TO | **0.25** (119s) |

Table 1: Modularity scores of Id, LE and FB on 30 directed graphs (TO = Time Out).

| Dataset | Size (#edges) | Id | Dh | LE | FB |
|---|---|---|---|---|---|
| AC | 58 595 | 0.01 (< 1s) | 0.46 (2s) | 0.07 (31s) | **0.72** (1s) |
| YG | 293 360 | 0.15 (< 1s) | 0.03 (5s) | 0.03 (98s) | **0.51** (2s) |
| GH | 440 237 | 0.21 (1s) | 0.07 (13s) | 0.07 (118s) | **0.33** (2s) |
| BX | 1 149 739 | 0.19 (3s) | 0.02 (28s) | 0.02 (893s) | **0.21** (5s) |
| ben | 1 164 576 | 0.02 (1s) | VE | 0.01 (319s) | **0.55** (2s) |
| SO | 1 301 942 | 0.02 (< 1s) | VE | 0.02 (502s) | **0.32** (6s) |
| TM | 1 366 466 | 0.04 (< 1s) | VE | TO | **0.61** (7s) |
| AM | 1 470 404 | 0.00 (3s) | 0.01 (18s) | 0.01 (184s) | **0.66** (5s) |
| Vut | 2 298 816 | 0.12 (1s) | 0.00 (1s) | 0.00 (15s) | **0.27** (1s) |
| Cut | 2 411 819 | 0.19 (1s) | 0.00 (3s) | 0.00 (96s) | **0.20** (2s) |
| But | 2 555 080 | 0.45 (1s) | 0.00 (2s) | 0.46 (34s) | **0.49** (1s) |
| DV | 3 018 197 | 0.02 (2s) | **0.10** (2s) | 0.02 (4s) | 0.08 (3s) |
| DBT | 3 232 134 | **0.04** (6s) | 0.01 (9s) | 0.01 (23s) | 0.01 (3s) |
| Wut | 4 664 605 | 0.04 (3s) | 0.00 (33s) | 0.00 (490s) | **0.45** (7s) |
| AR | 5 838 041 | 0.00 (4s) | 0.07 (516s) | TO | **0.39** (25s) |
| FG | 8 545 307 | 0.10 (3s) | 0.00 (28s) | TO | **0.20** (6s) |
| M3 | 10 000 054 | 0.06 (17s) | 0.09 (5s) | **0.11** (9s) | 0.09 (4s) |
| DI | 14 414 659 | 0.05 (2s) | 0.01 (177s) | TO | **0.32** (16s) |
| Ls | 19 150 868 | **0.12** (170s) | 0.05 (16s) | 0.03 (37s) | 0.04 (8s) |
| RE | 60 569 726 | 0.08 (19s) | 0.00 (25s) | 0.08 (141s) | **0.09** (22s) |

Table 2: Modularity scores of Id, DE, LE, FB on 20 bipartite graphs (VE = Value Error, TO = Time Out).

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
