# OpenReview forum: "The Forward-Backward Embedding of Directed Graphs"
_ICLR.cc/2019/Conference_

### Official Review · AnonReviewer1 · 2018-10-26
**An interesting and valuable idea, but flawed in terms of execution and distinction/difference to previous works**

**Rating:** 4
**Confidence:** 4

**Review:**

# Summary of the paper

This paper proposes an embedding of directed graphs based on the SVD of a normalized adjacency matrix. This embedding is shown to be equivalent to the spectral embedding of a co-citation graph, which is more complex to calculate. Interestingly, the proposed approach does not require the *explicit* representation of this graph. Moreover, the paper also shows that distances of the embedded vectors are proportional to mean commute times of a forward--backward random walk in the original graph. A suite of experiments is run on graphs from KONECT.

# Review

This is a well-written paper, which I enjoyed reading. The extension of embeddings to the case of directed graphs is significant and warrants a detailed exploration.

The principal issues I see with this paper are as follows:

- The originality or scope of the contribution is not clear
- The experimental section is uncompelling
- Several relevant works appear to have been ignored

Overall, I like the way the paper treats the subject. In particular, I appreciate the fact that proofs are explained well; additionally, code is provided, which will increase reproducibility. This is uncommon and praiseworthy!

As for the originality of the paper, I find it hard to judge the scope of the contribution. The paper is extremely well written and employs a very pedagogical treatment of the subject, which I appreciate. Yet, it is hard for me to judge the utility and novelty of the proposed method in light of Section 8, where the paper shows that a spectral embedding of the undirected variant of the graph leads to essentially the *same* eigenvectors (up to renormalization and permutations). To prove that a new method is more effective, this point should be emphasized more:

1. In a sense, I would see the results from Section 8 as the equal to what 'Laplacian Eigenmaps' (LE) yields. This needs to be stressed, and analysed in an experimental section.

2. I understand that the order of the singular vectors is different, so embeddings that use only parts of them will be different. However, a convincing experiment should assess the differences. For example, in which regime for $d$ (number of used vectors) will the new method be surpassed by the old one? Is there such a regime? Ideally, this will be answered in the form of an asymptotic theorem; it could also be a larger experiment, though (to simulate the conditions in practice).

3. I understand that the new approach has a lower run-time, because the SVD is more efficient than eigendecomposition. However, what about a simple baseline algorithm that uses SVD for the *undirected* graph of the input data? This should be simple to accomplish, and would be a way to ascertain the benefit of using edge directions.

  To my understanding, LE should be this embedding, but from the table, I can see that its runtime is a lot worse than the novel method. What causes this? The fact that eigendecomposition is used instead of SVD?  If so, an additional SVD-based approach should be implemented.

This brings me to the experimental section. Here, the paper demonstrates the superiority of the new embedding based on evaluating modularity of a set of different clusterings of larger graphs, obtained using $k$-means. I have several concerns about this:

1. Modularity has problems with larger networks because only a small part of the network will be used in its configuration.

2. Since the embeddings cannot be easily compared due to missing ground truth information, other metrics should be employed. Here are a few, which are often used  by the community. See 'Is there a best quality metric for graph clusters?' by Almeida et al. for more details and a description of their shortcomings:
    - Silhouette coefficient
    - Coverage
    - Conductance

  Different ones should be evaluated here in order to show the behaviour of the new embedding. Do the embeddings differ if the modularities are similar?

3. How do the results change for different values of $d$? I find it hard to disentangle such a discussion from instabilities in $k$-means, but to my understanding of the method, tuning $d$ means that more or less information is used from the singular vectors.

   This could also be quantified in a proof (about asymptotic behaviour) but an experiment would be equally fine.

Concerning the bibliography, or the treatment of prior works, there are some issues:

- There appear to be some missing references of earlier works that used SVD or variants in order to cluster graphs or embed them:

  - Drineas et al.: 'Clustering Large Graphs via the Singular Value Decomposition'
  - Malliaros and Vazirgiannis: 'Clustering and Community Detection in Directed Networks: A Survey'

- Likewise, the use of pseudo-inverse Laplacians has a lot more papers attached to it (these are only a few that are relevant):

  - Ho and Dooren: 'On the pseudo-inverse of the Laplacian of a bipartite graph'
  - Gutman and Xia: 'Generalized inverse of the Laplacian matrix and some applications'

# Suggestions for improvement

- In some sense, this work can be seen as an extension of Laplacian eigenmaps to the directed case. The paper needs to be more clear about these extensions with respect to prior work. In Section 2, it is claimed that 'our main contribution is a proper normalization'. This strikes me as a rather small contribution in light of the experimental section, as outlined above.

- I am also hesitant to speak about a better interpretability of the mean commute time. I agree that it is nice to know that the distance permits such an interpretation in terms of random walks, but what is the impact of knowing the MCT? It is not only used in the embedding insofar as one obtains a vector representation.

- Section 5 is then the standard way of defining random walks based on a Laplacian matrix, and the correspondence to the pseudo-inverse of the Laplacian is shown. This is mathematically interesting, but appears to me to be in line with previous research.

- The section about co-citation graphs should make it more clear that 'successors' are to be taken in terms of the original graph and the directionality of edges. Since this is a standard definition in the domain of network analysis I would suggest citing a textbook here.

- In Section 6, the paper could give more details about random walk concepts such as 'stochastic', 'stationary distribution' etc., as it would make the paper more accessible (I am familiar with these concepts but since the writing of the paper is of high quality in the other sections, I am convinced this would improve its impact, and attract more readers).

Typos & grammar issues:

- 'in terms of random walk' --> 'in terms of random walks'
- 'equivalent to build' --> 'equivalent to building'
- 'with corresponding unitary matrix' --> 'with a corresponding unitary
  matrix'
- 'square Euclidean distance' --> 'squared Euclidean distance'
- 'equivalent to consider' --> 'equivalent to considering'
- 'irreductible' --> 'irreducible'
- 'and provide generally' --> 'and provides generally'
- 'in low dimension' --> 'in a lower dimension'

Furthermore, the bibliography should employ consistent capitalization and journal names for articles.

---

> ### Public Comment · (anonymous) · 2018-11-23
> **Originality  and literature**
>
> We thank the reviewer for her/his valuable comments.
>
> The originality of our work lies in the interpretation of our embedding in terms of random walks, extending the known theoretical results for undirected graphs to directed graphs. We would like to emphasize the fact that our approach differs from the usual spectral embedding techniques in that the latter only applies to undirected graphs. The purpose of Section 8  is to show that, even in the case of undirected graphs, both methods differ (because the leading singular / eigenvectors are not the same); we agree that more work is needed to understand the exact connection between both approaches for undirected graphs, but this goes beyond the scope of the present paper, whose focus is on directed graphs.
>
> We thank the reviewer for the references that we will add to the paper; observe that none of them makes the connection between SVD and random walks.
>
> We agree with the reviewer that the choice of the quality metric is (always) questionable (hence the need for interpretable embedding techniques :-). However, the modularity is a fairly standard metric in the literature. Other metrics can easily be tested using the notebook made available. We plan to include experiments with labelled data (using the 20newsgroup dataset) if the paper is accepted.
>
> We thank the reviewer for the various suggestions. In particular, we will definitely highlight the contributions of the paper compared to existing work.
>
> The authors

---

> > ### Comment · AnonReviewer1 · 2018-11-26
> > **re: Originality and literature**
> >
> > Thanks for the reply! I agree with your assessment concerning undirected graphs. It would strengthen your paper immensely if you had a scenario where the improvements over the 'standard' approach are clearly shown (this does not necessarily have to be something complex, though!).
> >
> > I also agree with your thoughts about the quality metric. However, would it not be easier to have a metric that can be calculated in the original graph and in the embedding? Similar to quality metrics for dimensionality reduction (stress/strain, mean relative rank error, etc...), I wonder whether it would not be easier to check certain invariants prior and after embedding. Do you have any thoughts in this direction?

---

### Official Review · AnonReviewer3 · 2018-11-06
**Incremental and trivial observations**

**Rating:** 3
**Confidence:** 5

**Review:**

The paper studies embeddings of directed graphs based on SVD. It proposes an interpretation of the embedding obtained from the normalized adjacency matrix in terms of the forward-backward random walk on the graph. In such random walk odd steps are taken using the edges of the graph while even steps are taken using reverse edges. Such an interpretation seems to be a trivial extension of the work for undirected graphs.

For example, consider bipartite graphs, in such graphs the forward-backward random walk can be seen as a standard random walk on the graph with directions removed. Indeed, if the walk starts in part A then remove all edges from B to A and make A to B edges undirected.

---

> ### Public Comment · (anonymous) · 2018-11-06
> **Reply to the reviewer**
>
> Our paper is not a trivial extension of the work for undirected graphs. In particular, for bipartite graphs, the considered random walk is *not* a regular random walk as claimed in the review, but a two-hop random walk. To our knowledge, this has never been observed before. More generally, our paper is the first to relate SVD to random walks in directed graphs. This is very well known for spectral embedding, *not* for the SVD.

---

### Official Review · AnonReviewer2 · 2018-11-08
**well written - standard / light content**

**Rating:** 5
**Confidence:** 5

**Review:**

The paper discusses how to embed nodes of a graph to a vector space using singular value decomposition of the normalized adjacency matrix. The material in this paper are standard literature in graph theory and spectral analysis. The paper contains a well-written literature review. The contributions and not new enough for a publication.

---

> ### Public Comment · (anonymous) · 2018-11-23
> **SVD and random walks**
>
> We thank the reviewer for her/his comments.
>
> We would like to emphasize the fact that our paper is the first to link two key techniques for graph embedding: SVD and random walks. While SVD is a standard tool in graph analysis, the resulting embedding techniques for graphs can be interpreted in terms of random walks only after proper normalization, as shown in the paper. One of the major challenges for research in machine learning is to make algorithms / decisions interpretable, and we believe that the proper embedding of graph data is a valuable step in this direction.
>
> The authors

---

### Public Comment · ~Michael_Bronstein1 · 2018-10-07
**related methods?**

Interesting work - one of a few to explicitly address directed graphs. I wonder if and how your approach related to the following two recent works:

1. MotifNet: a motif-based Graph Convolutional Network for directed graphs, arXiv:1802.01572 - using sub-graph structures (motifs) to convert directed graphs into (re)-weighted undirected graphs. This is the closest analogy of anisotropic diffusion on graphs.

2. Dual-Primal Graph Convolutional Networks, arXiv:1806.00770 - using convolutions on the line graph to update the edge weights; also can be considered as a generalization of the graph attention (GAT) approach.

---

> ### Public Comment · (anonymous) · 2018-10-08
> **Reply about related methods**
>
> Thank you for the interest in our work and for the references.
>
> > I wonder if and how your approach related to the following two recent works:
> > 1. MotifNet: a motif-based Graph Convolutional Network for directed graphs, arXiv:1802.01572 - using sub-graph structures (motifs) to convert directed graphs into (re)-weighted undirected graphs. This is the closest analogy of anisotropic diffusion on graphs.
>
> Our approach also relies on the conversion of a directed graph into a weighted undirected graph (the normalized co-citation graph). The key point is that this conversion is implicit, i.e., we do not need to actually construct this graph. The diffusion in the directed graph occurs through a forward-backward random walk.
>
> As far as we understand, the conversion we use is not in the class of motif-based graph conversions. The appropriate motif would be motif 10 of Fig. 2, with nodes i,j pointing to k, but we would need to:
> * count each such motif whenever it corresponds to a subset of the edges (not necessaritly a subgraph)
> * normalize this count by the degree of k.
>
> So we believe our work is only loosely related to this paper.
>
> > 2. Dual-Primal Graph Convolutional Networks, arXiv:1806.00770 - using convolutions on the line graph to update the edge weights; also can be considered as a generalization of the graph attention (GAT) approach.
>
> We do not see any direct link with this work. The considered normalized co-citation graph is not a line graph. In particular, the set of vertices of the normalized  co-citation graph is that of the original graph, without the sinks. It is not the set of edges.
>
> The authors

---

### Meta-Review · Area_Chair1 · 2018-12-13
**Unconvincing originality**

**Confidence:** 5
**Recommendation:** Reject

**Metareview:**

The reviewers are unanimous in their assessment that the paper lacks originality in its current form to be publishable at ICLR-2018.